

# Genome-wide identification and classification of the *Hsf* and *sHsp* gene families in *Prunus mume*, and transcriptional analysis under heat stress

Xueli Wan[1,2], Jie Yang[1,3], Cong Guo[1,4], Manzhu Bao[1] and Junwei Zhang[1]

[1] Key Laboratory of Horticultural Plant Biology, Ministry of Education, College of Horticulture and Forestry Sciences, Huazhong Agricultural University, Wuhan, China
[2] College of Landscape and Forestry, Qingdao Agricultural University, Qingdao, China
[3] School of Nuclear Technology and Chemisty & Biology, Hubei University of Science and Technology, Xianning, China
[4] Institute of Industrial Crops, Hubei Academy of Agricultural Sciences, Wuhan, China

## ABSTRACT

The transcriptional activation of heat shock proteins (Hsps) by heat shock transcription factors (Hsfs) is presumed to have a pivotal role in plant heat stress (HS) response. *Prunus mume* is an ornamental woody plant with distinctive features, including rich varieties and colors. In this study, 18 Hsfs and 24 small Hsps (sHsps) were identified in *P. mume*. Their chromosomal locations, protein domains, conserved motifs, phylogenetic relationships, and exon–intron structures were analyzed and compared with *Arabidopsis thaliana* Hsfs or sHsps. A total of 18 PmHsf members were classified into three major classes, A, B, and C. A total of 24 PmsHsps were grouped into eight subfamilies (CI to CIII, P, endoplasmic reticulum, M, and CI- or P-related). Quantitative reverse transcription PCR analysis revealed that members of the A2, A7, and A9 groups became the prominent Hsfs after heat shock, suggesting their involvement in a key regulatory role of heat tolerance. Most of the *PmsHsp* genes were up-regulated upon exposure to HS. Overall, our data contribute to an improved understanding of the complexity of the *P. mume Hsf* and *sHsp* gene families, and provide a basis for directing future systematic studies investigating the roles of the *Hsf* and *sHsp* gene families.

## INTRODUCTION

As sessile organisms, plants are unavoidably exposed to heat stress (HS). HS disturbs plant cellular homeostasis, which results in additional injuries. To relieve these adverse effects, higher plants have developed many responses to HS (*Mittler, Finka & Goloubinoff, 2012*). The accumulation of heat shock proteins (Hsps) is regulated by heat shock transcription factors (Hsfs), which play crucial roles in the HS response (HSR) and confer thermo-tolerance to plants and other organisms (*Kotak et al., 2007*; *Ohama et al., 2016*). Plant Hsfs contain at least two functional domains. All plant Hsfs share a DNA-binding

Corresponding author
Junwei Zhang,
zjw@mail.hzau.edu.cn

domain (DBD) at the N-terminus and an adjacent oligomerization domain (OD or HR-A/B), as well as nuclear localization signal (NLS) motifs (*Scharf et al., 2012*). The most conserved part of Hsfs is the DBD, which has a conformation containing a three-helix bundle and a four-stranded antiparallel β-sheet that ensures its specific interactions with HS promoter elements. The amino acid sequence of the OD contains a heptad repeat pattern of hydrophobic amino acid residues. Three plant Hsf classes are distinguished based on the number of amino acid residues between two heptad repeats. There are 21 amino acid residues in class A Hsfs, 7 in class C, and none in class B (*Nover et al., 2001*). Class A Hsfs function in transcription activation through AHA motifs, which are made up of aromatic, hydrophobic, and acidic amino acid residues, while these do not appear in class B and C Hsfs (*Döring et al., 2000*; *Kotak et al., 2004*). Additionally, there is a repressor domain with an LFGV motif in the C-terminal domain of Class B Hsfs.

Plant Hsps are grouped into five families based on their approximate molecular weights: Hsp100, Hsp90, Hsp70, Hsp60, and small Hsp (sHsp) (*Wang et al., 2004*). Most Hsp families (i.e., Hsp100, Hsp90, Hsp70, and Hsp60) are highly conserved across great organismal distance, and are among the most highly conserved protein families known (*Stechmann & Cavalier-Smith, 2003*; *Waters, Aevermann & Sanders-Reed, 2008*). Although the monomers of plant sHsp proteins are the smallest among Hsps (12–40 kDa), plant sHsps exhibit high diversity in amino acid sequence (*Hilton et al., 2013*; *Mani, Ramakrishna & Suguna, 2015*). Except for the conserved α-crystallin domain (ACD) near the C-terminus, the N- and C-terminal extensions are variable (*Kriehuber et al., 2010*; *Haslbeck & Vierling, 2015*). Despite the long evolutionary histories of many sHsp subfamilies, the sHsp family continues to expand within different angiosperm lineages (*Waters, Aevermann & Sanders-Reed, 2008*).

The ACD represents the conservative signature motif of sHsps, and consists of two antiparallel β-sheets (*Van Montfort et al., 2001*; *Van Montfort, Slingsby & Vierling, 2002*; *Delbecq & Klevit, 2013*). In addition to the diversity in their sequences, the plant sHsps are found in different subcellular locations, including the cytosol/nucleus (C), chloroplast (CP), endoplasmic reticulum (ER), mitochondria (MT), and peroxisome (PX). According to their cellular localization and sequence similarity, plant sHsps are fall into 11 subfamilies: subfamilies CI to CVI, subfamily CP, subfamily ER, subfamilies MTI and MTII, and subfamily PX (*Waters, 2013*).

Recently, Hsfs and sHsps have been identified from several higher plants. For example, 24 Hsfs have been identified in *Arabidopsis* (*Nover et al., 2001*), 24 in tomato (*Scharf et al., 2012*), 27 in poplar (*Zhang et al., 2015a*), 40 in cotton (*Wang et al., 2014*), 27 in willow (*Zhang et al., 2015a*), 25 in rice (*Scharf et al., 2012*), and 56 in wheat (*Xue et al., 2014*). Additionally, 19, 33, 37, 94, 23, and 27 sHsps have been identified in *Arabidopsis* (*Scharf, Siddique & Vierling, 2001*), tomato (*Arce et al., 2018*), poplar (*Zhang et al., 2015b*), cotton (*Ma et al., 2016*), rice (*Sarkar, Kim & Grover, 2009*), and wheat (*Pandey et al., 2015*), respectively. This reveals that the number of different Hsfs and sHsps varies among plant species, which may be the result of gene duplication and whole-genome duplication (WGD).

Under high temperature stress, plant Hsfs have two expression patterns: constitutive and induced expression. The mRNA levels of *Arabidopsis* HsfA2, B1, A4a, B2a, B2b, and A7a are significantly increased after HS, while the mRNA levels of the other Hsfs appear to be unchanged when comparing the control and HS conditions (*Busch, Wunderlich & Schöffl, 2005*). In maize, the transcription levels of five ZmHsfs do not change under HS, while 12 ZmHsfs are up-regulated (*Lin et al., 2011*). In contrast to the expression patterns of plant sHsfs, almost all plant sHsps are heat-inducible (*Waters, 2013*).

*Prunus mume* is an important ornamental plant in China. We previously detected the expression of *PmHSP17.9*, an sHsp of *P. mume*, under abiotic stresses, and its overexpression in *Arabidopsis* can improve heat tolerance and superoxide dismutase activity (*Wan et al., 2016*). In this study, we identified 18 members of the Hsf family and 24 members of the sHsp family based on the *P. mume* genome. Then, we conducted comprehensive analyses of the gene chromosomal locations, structures, phylogeny, and conserved motifs. To examine their potential roles, we investigated the expression patterns of the *Hsf* and *sHsp* genes in response to HS. The results of this study provide an overview of the *PmHsf* and *PmsHsp* genes, and lay the foundation for further functional analyses of these genes in response to HS.

## MATERIAL AND METHODS

### Identification and chromosomal locations of Hsf and sHsp members in *P. mume*

The genome, transcript, and protein sequences were accessed from the *P. mume* genome (V1.0) (https://www.ncbi.nlm.nih.gov/genome/?term=13911). To identify all potential Hsfs in *P. mume*, *Arabidopsis* Hsf protein sequences were retrieved from the *Arabidopsis* Information Resource (http://www.arabidopsis.org) and were used as queries to perform BLAST searches against the *P. mume* genome database to identify results with *e*-values less than 0.0001. To identify sHsp members, the *P. mume* genome annotation database was searched with the keyword "alpha crystallin protein." Then, the putative sHsp sequences were used as queries in NCBI BLAST searches. Next, the sequences were manually analyzed to exclude duplicated annotations in the NCBI database.

The locations of the Hsf and sHsp members on the *P. mume* chromosomes were obtained from the *P. mume* genome data. PmHsfs or PmsHsps were numbered (1, 2, 3, etc.) according to their order on the chromosomes. Finally, chromosome location graphics for the Hsfs and sHsps were generated using MapInspect software.

### Gene structure and conserved motif analysis

Exon–intron structure information was obtained from the *P. mume* genome database. The gene structures of the Hsf or sHsp members were generated on the Gene Structure Display Server 2.0 (http://gsds.cbi.pku.edu.cn). Then, the theoretical isoelectric points and molecular weights were estimated using the ExPASy software (http://web.expasy.org). Finally, the conserved motifs of the Hsf or sHsp proteins were analyzed by submitting their full-length amino acid sequences to the MEME 5.0.5 online tool (http://meme-suite.org/tools/meme) (*Bailey et al., 2006*). The parameters used were as follows: number of

repetitions: any; maximum number of motifs: 30 for Hsfs and 10 for sHsps; the other parameters used the default settings.

## Phylogenetic analysis

To further understand the evolutionary relationships of the Hsf and sHsp proteins, phylogenetic trees were constructed based on the complete protein sequences of the Hsfs or sHsps from *P. mume* using the MEGA 5.0 software with its default settings using the neighbor-joining method (*Saitou & Nei, 1987*). The evolutionary distances between the PmHsf or PmsHsp sequence pairs were computed using the ClustalW tool. Bootstrap value analysis was performed using 1,000 replicates to assess the level of statistical support at each node.

## Plant material and growth conditions

Dormant cuttings with five buds each were harvested from three 5-year-old *P. mume* "Xue Mei" trees at the Huazhong Agricultural University (Wuhan, China). Next, the lower ends of the cuttings were inserted in 20 mL of distilled water in a phytotron with a 14-h photoperiod, a light intensity of 120 µmol $m^{-2}s^{-1}$, and a temperature of 24 °C until leaf expansion of one to two cm length occurred. Cuttings with well-expanded shoots were transferred into a chamber and subjected to 42 °C HS. The leaves were harvested after 1, 2, 4, 8, and 12 h of HS, and leaves in the phytotron at 24 °C were used as a negative control. All samples were frozen immediately in liquid nitrogen and stored at −80 °C until RNA extraction.

## RNA isolation and quantitative reverse transcription PCR analysis

Total RNA was extracted using the EASYspin Rapid Plant RNA Extraction Kit (RN09; Aidlab Biotechnologies, Beijing, China), and RNase-free DNase I (Qiagen, Valencia, CA, USA) was used to remove any contamination of genomic DNA according to the manufacturer's protocol. First-strand cDNA synthesis was carried out with one µg of total RNA using the PrimeScript® RT Reagent Kit (DRR047A; TaKaRa, Shiga, Japan).

Gene-specific oligonucleotide primers (Table S1) were designed using the Primer Premier 5 software. Each qRT-PCR primer pair was tested by end-point PCR to show that they produced a specific product of predicted size. Primer efficiencies were not calculated. Water was used as non-template control to detect the presence of any contaminating DNA. qRT-PCR was performed using the ABI 7500 Fast Real-Time PCR System (Applied Biosystems, Foster City, CA, USA) using SYBR® *Premix Ex Taq*™ II (Tli RNaseH Plus; Takara, Dalian, China). The PCR cycling protocol was as follows: 95 °C for 30 s followed by 40 cycles of 95 °C for 3 s and 60 °C for 30 s. The transcriptional expression levels were determined using the $2^{-\Delta\Delta CT}$ method (*Schmittgen & Livak, 2008*). The *eukaryotic translational elongation factor 1 alpha* (*PmEF1α*) gene of *P. mume* (previously published in the past study) was used as the reference gene (*Guo et al., 2014*). Three biological and technical replicates for each reaction were performed. A total of 0.5 cycles range in Cq value was acceptable within a single technical triplicate. Statistical significance was determined by one-way analysis of variance using SPSS ver. 19.0 software (IBM Corp., Armonk, NY, USA). Differences were considered to be significant at $p < 0.05$.

## RESULTS

### Identification and chromosomal locations of the *Hsf* and *sHsp* genes in *P. mume*

Through automated database searching and manual checks, 18 nonredundant Hsf and 24 sHsp genes were identified from the *P. mume* genome database. Detailed information, such as gene IDs, coding sequence length, number of amino acids, and molecular weight, is provided in Table S2, Figs. S1 and S2.

To determine their distribution in the *P. mume* genome, the locations of the *Hsf* and *sHsp* genes were mapped onto the *P. mume* chromosomes, and the genes were designated as *PmHsf1-18* and *PmsHsp1-24* based on the order of their locations on the chromosomes (Fig. 1). *PmHsf18* was anchored on un-assembled scaffold272, whereas all of the remaining *Hsf* genes were distributed on chromosomes 2–8. The numbers of *Hsf* genes on each chromosome were uneven: chromosomes 2, 8, 3, and 7 contained five, four, three, and two *Hsf* genes, respectively; and chromosomes 4, 5, and 6 harbored only one *Hsf* gene each (Fig. 1A). *PmsHsp23* and *PmsHsp24* were located at scaffold56 and scaffold265, respectively. Furthermore, no *sHsp* genes were detected on chromosome 6, whereas the highest number of *sHsp* genes was located on chromosome 1 (Fig. 1B).

### Phylogenetic analysis of the Hsf and sHsp proteins from *P. mume* and *Arabidopsis*

To evaluate the phylogenetic relationships of the Hsf or sHsp proteins in depth, phylogenetic trees were constructed based on the full-length amino acid sequences of the proteins from *P. mume* and *Arabidopsis* (Figs. 2 and 3). The PmHsf family clearly grouped into three major classes (A, B, and C). Class A contained 11 Hsf genes, and was further subdivided into nine distinct subfamilies (A1–A9). Class B contained six Hsfs, and was classified into four subfamilies (B1–B4). Class C Hsf constituted one distinct clade, which appeared more closely related to the Hsf A group (Fig. 2).

The phylogenetic analysis revealed that the sHsp family could be classified into cytoplasmic/nuclear (CI, CII, and CIII), plastid (P), ER, mitochondrial (M), CI(r) (CI-related), and P(r) (P-related) subfamilies in *P. mume*. However, there were some differences between the phylogenetic analysis and the annotations. For example, PmsHsp23 (annotated as M) grouped into the P(r) subfamily. Additionally, PmsHsp10 and PmsHsp16 (annotated as class CI) grouped into the ER subfamily, whereas PmHsp15 (annotated as CV) and PmHsp24 (annotated as CVI) grouped into the CI(r) subfamily. Notably, many sHsps grouped to class CI, CI(r), CII, and CIII, located in the cytosol/nucleus, suggesting that the cytosol may be the main functional location for PmsHsps (Fig. 3).

### Analyses of gene structure, conserved domains, and motifs

In an effort to gain deeper insights into the structural differences between the *Hsf* and *sHsp* genes in *P. mume*, the exon–intron structures of *Hsf* and *sHsp* genes were analyzed (Figs. 2 and 3; Table S2). All of the *Hsf* genes contained only one intron, which had a position that was highly conserved, except for *PmHsf2*, which contained two introns. The length of the introns varied among the *Hsf* genes: for example, *PmHsf12* and *PmHsf17*

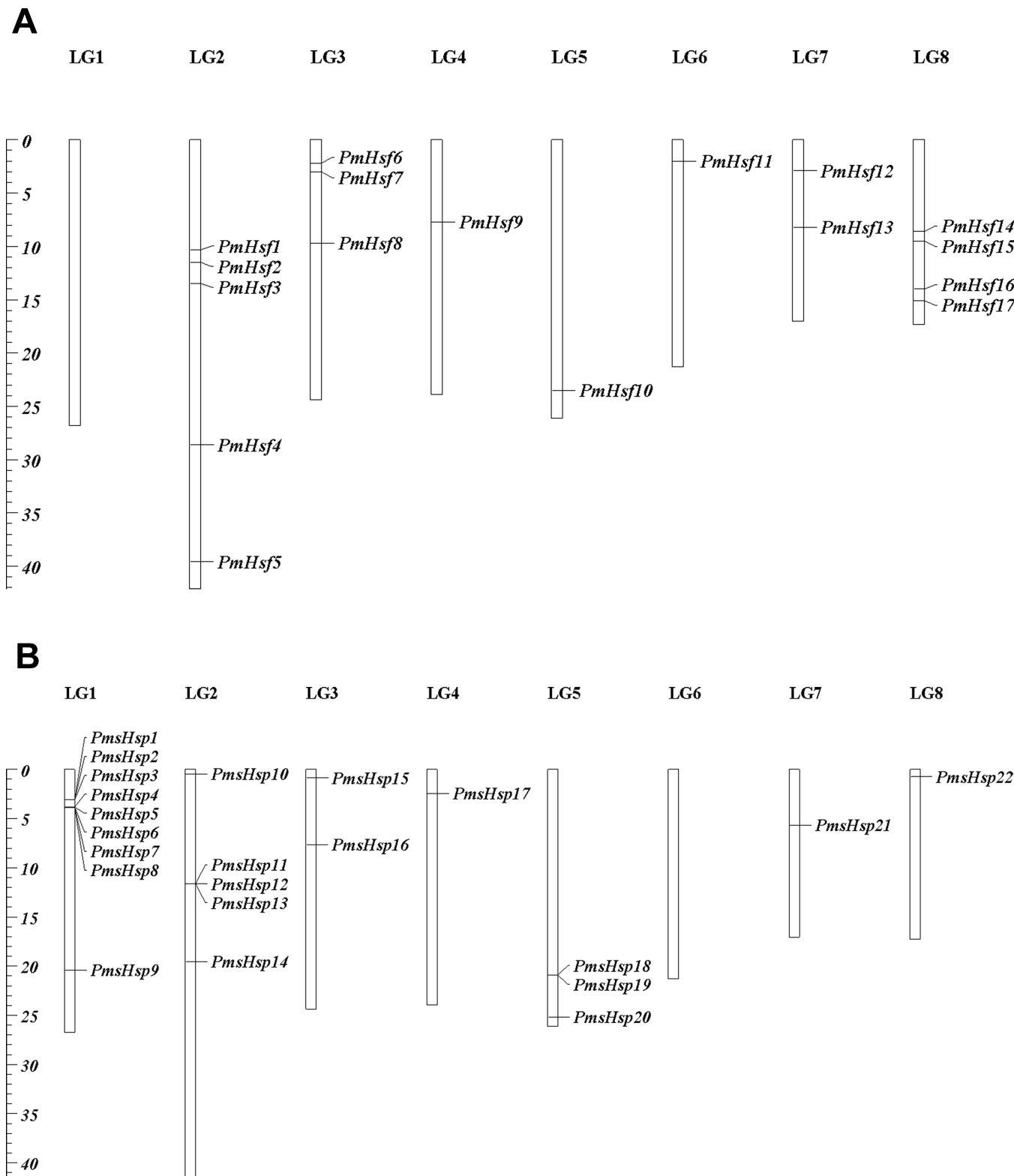

**Figure 1 Locations of *Hsf* and *sHsp* genes on the *Prunus mume* chromosomes.** (A) Locations of 18 *Hsf* genes on the *P. mume* chromosomes. (B) Locations of 24 *sHsp* genes on the *P. mume* chromosomes. The scale represents megabases (Mb). The chromosome numbers are indicated at the top of each bar.

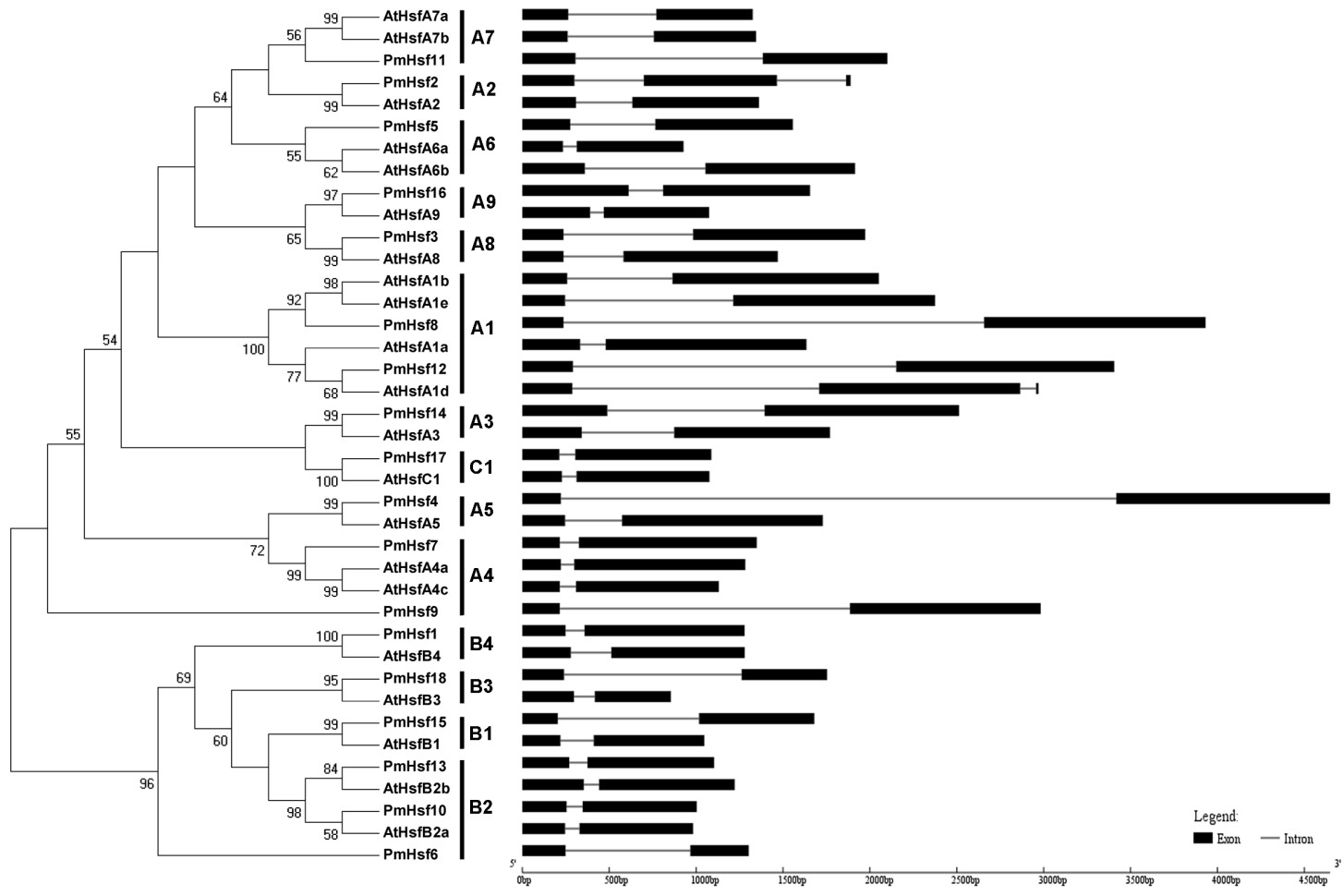

**Figure 2 Phylogenetic trees and exon-intron structures for *PmHsf* and *Arabidopsis Hsf* families.** The phylogenetic trees were obtained using the MEGA 5.0 software on the basis of the complete protein sequences. Bootstrap values > 50 are shown. Exons are indicated by black boxes. Introns are represented by black lines.

had the smallest intron (92 bp), and *PmHsf4* had the longest intron (3,196 bp). However, the number and location of the introns varied among the *sHsp* genes (Fig. 3). Most *sHsp* genes contained no introns or only one intron, whereas *PmsHsp18* contained two introns. The largest number of introns was found in *PmsHsp2*, which contained four introns (Fig. 3).

The known elements of the functional domains of AtHsf allowed for the study of similar domains in the 18 PmHsfs (Table 1). Five conserved domains (DBD, HR-A/B region, NLS, nuclear export signal (NES), and AHA motifs) were observed in most of the PmHsf proteins. Multiple alignments revealed that the highly conserved DBD of approximately 100 amino acids was located close to the N-terminus in all of the PmHsfs (Table 1; Fig. 4A). Interestingly, PmHsf14 and PmHsf16 contained a long N-terminus (102 and 143 amino acids, respectively), whereas most Hsfs had a short N-terminus (Fig. S1). The HR-A/B regions of the PmHsfs were consistently observed with the predicted coiled-coil structure (Table 1; Fig. 4B). Two clusters of basic amino acid residues (K/R), which are found in nearly all PmHsf proteins, may contribute to the potential NLS motif

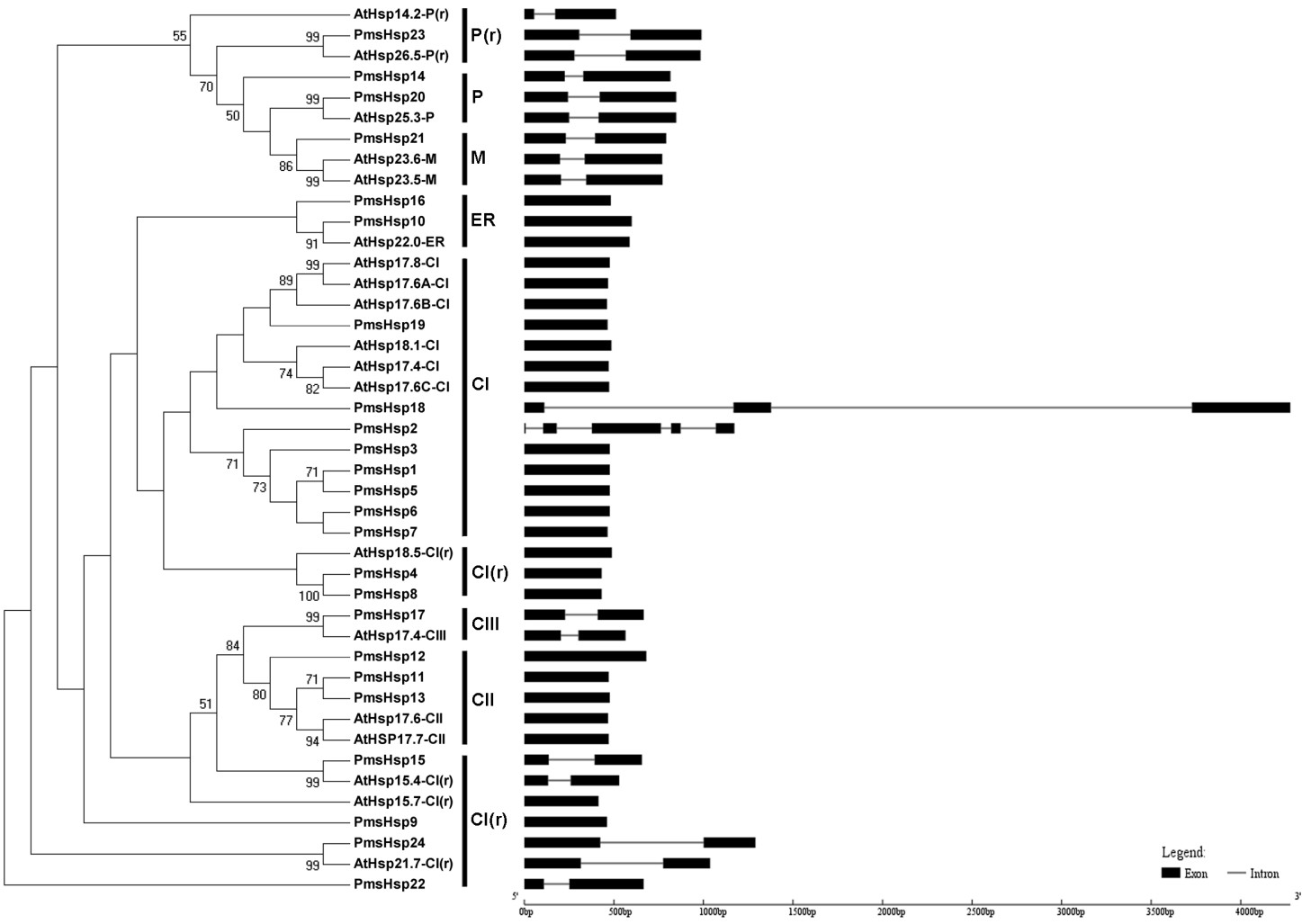

**Figure 3 Phylogenetic trees and exon-intron structures for *PmsHsp* and *Arabidopsis sHsp* families.** The phylogenetic trees were obtained using the MEGA 5.0 software on the basis of the complete protein sequences. Bootstrap values > 50 are shown. C, cytoplasmic/nuclear; ER, endoplasmic reticulum; P, plastids; M, mitochondria; CI(r), CI related; P(r), P related. Exons are indicated by black boxes. Introns are represented by black lines.

(*Lyck et al., 1997*). As expected, most Hsf proteins contained the putative NES (Table 1), composed of hydrophobic, frequently leucine-rich amino acid residues (*La Cour et al., 2004*). Meanwhile, the putative AHA motifs were located in the center of the CTAD for most class A PmHsfs, with variable lengths and richness in F, W, D, and L amino acid residues (Table 1).

The conserved motif distributions were searched using the MEME web server to obtain more insights into the predicted domains and the diversity of the motif compositions (Fig. 5; Table 2). Motifs 1, 2, and 3 represented the Hsf DBD domains (Table 2). Motif 4 indicated that the HR-A/B region and was found in all Hsfs, whereas class A proteins also exhibited the motif 5-type HR-A/B region. Motifs 9 and 22 represented the NLS domain, which was widely distributed in the PmHsf family. Furthermore, motifs 10 and 6 represented NES and AHA motifs, respectively. Overall, the MEME motifs indicated that the PmHsfs contained putative conserved domains.

**Table 1 Functional domains of non-redundant Hsfs in *P. mume*.**

| Name | Type | DBD | Linker (bp) | HR-A/B | NLS | NES | AHA motifs |
|------|------|-----|-------------|--------|-----|-----|-----------|
| PmHsf8 | HsfA1 | 18–111 | 36 | 148–198 | (215) NRR 5 KKRRLPR | (489) ITEQMELL | (439) DIFWEQFLPA |
| PmHsf12 | HsfA1 | 36–129 | 36 | 166–216 | (233) NRR 5 KKRRLK | (500) LTEKMERL | (457) WDQFLQ |
| PmHsf2 | HsfA2 | 39–132 | 29 | 162–212 | (223) EKKARNKE 7 RKRR | | (279) ETFFSAAALD (315) WEELWSDE |
| PmHsf14 | HsfA3 | 102–195 | 24 | 220–270 | (292) RMKRK | (322) AWRNLS | (366) DVASDELNLS (493) DIGPLHAAG |
| PmHsf7 | HsfA4 | 11–104 | 33 | 138–188 | (206) RKRRLPR | (399) LTEQMGHL | (250) LTFWEDTIFD (348) DVFWEHFLTE |
| PmHsf9 | HsfA4 | 11–104 | 31 | 138–186 | (203) NKKRR | (426) FTNQIGRL | (252) LNFWEDFLHG (376) DMFWEQCLTE |
| PmHsf4 | HsfA5 | 13–106 | 26 | 133–183 | (206) KKRR | (478) AETLTL | (431) DVFWEQFLTE |
| PmHsf5 | HsfA6 | 31–124 | 29 | 154–204 | (217) KDKRN 7 KKRRR | (343) FVEELVYL | (293) DEEECMEKEEGN (336) EDEDVDV |
| PmHsf11 | HsfA7 | 41–134 | 29 | 164–214 | (227) KDKRK 7 KKRRR | (328) LADRLGYL | (302) DEESERFEGDL (321) EDEDVII |
| PmHsf3 | HsfA8 | 18–111 | 35 | 147–197 | (328) KEDGK | (387) ITEQMGYL | (308) DGAWEQLLLA (350) ESQNFDTLIE |
| PmHsf16 | HsfA9 | 143–235 | 25 | 261–311 | (323) KRVIKR 7 RKRRR | | (355) DTSLSVDCG |
| PmHsf15 | HsfB1 | 7–100 | 61 | 162–191 | (245) DEKKK | | |
| PmHsf6 | HsfB2 | 16–115 | 40 | 156–185 | | | |
| PmHsf10 | HsfB2 | 24–117 | 56 | 174–203 | (248) KRARE | | |
| PmHsf13 | HsfB2 | 29–122 | 69 | 192–221 | (291) KRVRR | | |
| PmHsf18 | HsfB3 | 19–112 | 52 | 165–194 | (180) KRKCK | (208) PKLFGVRL | |
| PmHsf1 | HsfB4 | 22–115 | 96 | 212–241 | (335) KNTK 9 KKR | (379) LGLHLM | |
| PmHsf17 | HsfC1 | 9–102 | 35 | 138–174 | (198) KKRR | | |

The multiple sequence alignments revealed that the conserved ACD, which comprised eight β-strands and two conserved domains (consensus region I and II) specific for sHsps, was located in the C-terminal region of the transcript of the *PmsHsp*s (Fig. S2; Fig. 6). The "GVL" residues, which are highly conserved in consensus region I, were substituted in PmsHsp9, PmsHsp22, and PmsHsp24. In contrast, the C-terminal portions of the PmsHsps were quite variable in both sequence and length (Fig. S2).

The distributions of the conserved motifs in the PmsHsp proteins were identified using the MEME web server, and ten distinct motifs were discovered (Fig. 7). The PmsHsps shared similar motif composition and order (Table 3). The motifs 1, 2, 3, 4, and 8 represented the ACD, which are widely distributed in the PmsHsp family. The conserved motifs 5 and 7 were distributed mainly in the N-terminal regions and motif 6 was distributed in the C-terminal regions, representing secondary structural element β10. However, the biological significance of these motifs requires further examination.

## Expression patterns of the putative *PmHsf* and *PmsHsp* genes under HS

To assess the roles of the predicted PmHsf and PmsHsp genes in the process of HS, qRT-PCR was performed to detect the transcriptional profile for 18 *PmHsf* and 24 *PmsHsp*

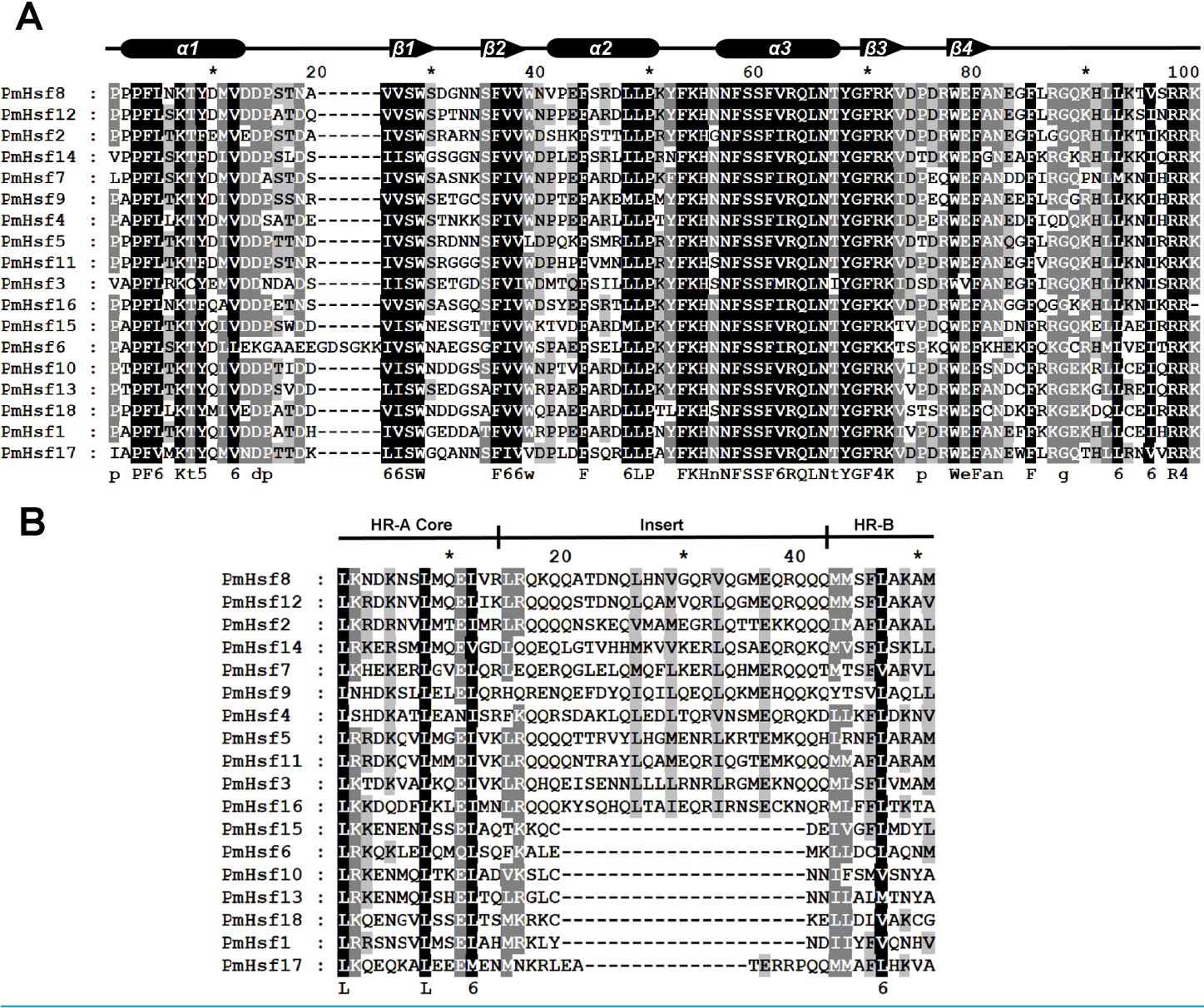

**Figure 4 Multiple sequence alignment of the DBD domains and HR-A/B regions of the Hsf proteins in *P. mume*.** (A) Multiple sequence alignment of the DBD domains of the Hsf proteins in *P. mume*. The multiple alignment results clearly show the highly conserved DBD domains among *P. mume Hsf* genes. The secondary structure elements of DBD (α1-β1-β2-α2-α3-β3-β4) are shown above the alignment. Cylindrical tubes represent α-helices and block arrows represent β-sheets. (B) Multiple sequence alignment of the HR-A B regions of the Hsf proteins in *P. mume*. The scheme at the top depicts the locations and boundaries of the HR-A core, insert, and HR-B regions within the HR-A/B regions. The structures between HR-A and HR-B consist of 21 amino acid and seven amino acid insertions, respectively, for Class A and C.



genes in the leaves of *P. mume* exposed to HS. The results revealed that these genes were differentially expressed in response to HS (Figs. 8 and 9).

The transcriptional levels of *PmHsf3* (A8), *PmHsf4* (A5), *PmHsf8* (A1), *PmHsf9* (A4), and *PmHsf12* (A1) did not change at most of the HS time points and were down-regulated at some time points (control, 1, 2, 4, 8, and 12 h at 42 °C), whereas the expression of *PmHsf18* (B3) was inhibited after HS. In contrast to those genes, the transcript abundances

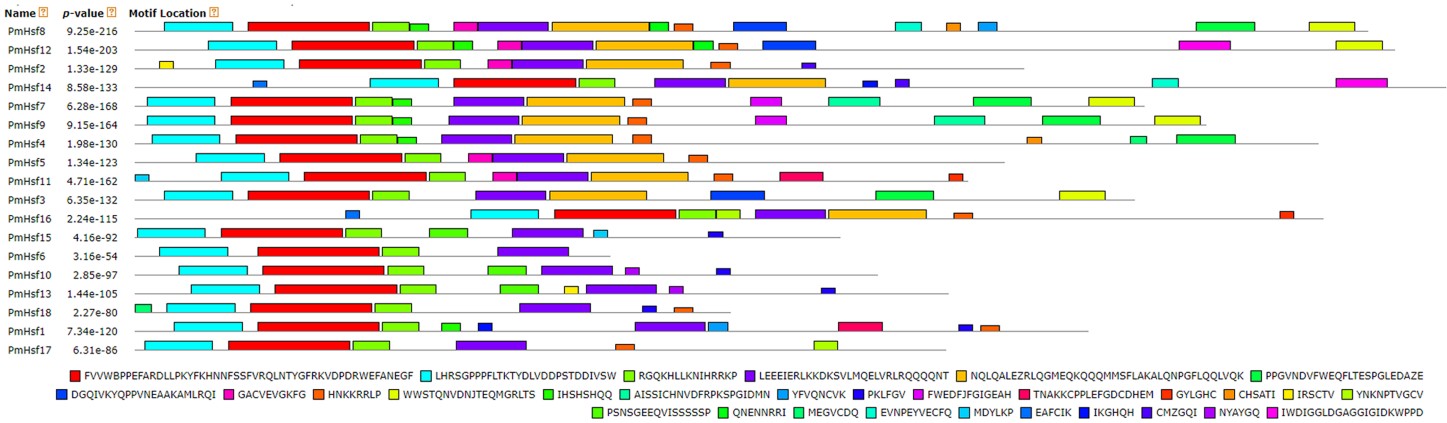

**Figure 5 Distribution of conserved motifs in the *Hsf* family members in *P. mume*.** All motifs were identified by MEME 5.0.5 using the complete amino acid sequences. Motif sizes are indicated at the bottom of the figure. Different motifs are indicated by different colors numbered 1–30. For details of motifs refer to Table 2.

revealed that the remaining genes were up-regulated and their expression patterns were assigned to two categories. The expression levels of *PmHsf1* (*B4*), *PmHsf5* (*A6*), *PmHsf7* (*A4*), and *PmHsf17* (*C1*) slowly increased and reached the highest levels at 8 or 12 h of HS. The other eight genes, *PmHsf6* (*B2*), *PmHsf10* (*B2*), *PmHsf13* (*B2*), *PmHsf2* (*A2*), *PmHsf11* (*A7*), *PmHsf14* (*A3*), *PmHsf15* (*B1*), and *PmHsf16* (*A9*), exhibited enhanced expression immediately after HS, and the mRNA levels of these genes slowly or quickly decreased in the later period following HS treatment (Fig. 8).

Under HS, the expression of most of the *PmsHsps* increased, except for *PmsHsp22*. The transcripts of 13 of the up-regulated *PmsHsps* were present at a high abundance, that is, *PmsHsp1\**, *4\**, *10*, *11*, *13*, *14*, *16*, *17*, *18–21*, and *23*, which exhibited the highest expression at 1 h of HS, followed by a rapid or gradual fall, except for *PmsHsp18*; meanwhile, *PmsHsp12* and *15* were expressed at moderately high levels, and the mRNA levels of *PmsHsp9* and *24* were low (Fig. 9).

## DISCUSSION

### Identification and conserved motif analysis of *Hsf* and *sHsp* genes from the *P. mume* genome

An increasing number of studies indicate that Hsf and sHsp proteins play important roles in the adaptation to environmental stress. However, no specific information is available regarding the *Hsf* and *sHsp* genes in *P. mume*. In this study, we conducted a comprehensive analysis of the PmHsf and PmsHsp families. Plant genomes contain numerous *Hsf* genes, with varying members among species (*Nover et al., 2001*; *Baniwal et al., 2004*; *Lin et al., 2011*; *Xue et al., 2014*). In four species of the Rosaceae family, *Malus domestica*, *Fragaria vesca*, *Pyrus bretschneideri*, and *Prunus persica*, 25, 17, 29, and 17 *Hsfs* were identified, respectively (*Giorno et al., 2012*; *Hu et al., 2015*; *Qiao et al., 2015*). In our study, we used AtHsf protein sequences as queries, which resulted in the identification of 18 *Hsfs* from the available *P. mume* genomic data. The numbers of *Hsf* genes in pear and apple were nearly two times greater than those in peach, strawberry, and Mei flower

**Table 2 Hsf motif sequences identified in *P. mume* by MEME tools.**

| Motif | Multilevel consensus sequence |
| --- | --- |
| 1 | FVVWBPPEFARDLLPKYFKHNNFSSFVRQLNTYGFRKVDPDRWEFANEGF |
| 2 | LHRSGPPPFLTKTYDLVDDPSTDDIVSW |
| 3 | RGQKHLLKNIHRRKP |
| 4 | LEEEIERLKKDKSVLMQELVRLRQQQQNT |
| 5 | NQLQALEZRLQGMEQKQQQMMSFLAKALQNPGFLQQLVQK |
| 6 | PPGVNDVFWEQFLTESPGLEDAZE |
| 7 | DGQIVKYQPPVNEAAKAMLRQI |
| 8 | GACVEVGKFG |
| 9 | HNKKRRLP |
| 10 | WWSTQNVDNJTEQMGRLTS |
| 11 | IHSHSHQQ |
| 12 | AISSICHNVDFRPKSPGIDMN |
| 13 | YFVQNCVK |
| 14 | PKLFGV |
| 15 | FWEDFJFGIGEAH |
| 16 | TNAKKCPPLEFGDCDHEM |
| 17 | GYLGHC |
| 18 | CHSATI |
| 19 | IRSCTV |
| 20 | YNKNPTVGCV |
| 21 | PSNSGEEQVISSSSSP |
| 22 | QNENNRRI |
| 23 | MEGVCDQ |
| 24 | EVNPEYVECFQ |
| 25 | MDYLKP |
| 26 | EAFCIK |
| 27 | IKGHQH |
| 28 | CMZGQI |
| 29 | NYAYGQ |
| 30 | IWDIGGLDGAGGIGIDKWPPD |

**Note:**
Numbers correspond to the motifs described in Fig. 5.

(Fig. S3), which may be the result of a WGD event in the Maloideae. A similar situation was observed in *Populus trichocarpa* and *Zea mays* (*Lin et al., 2011*; *Zhang et al., 2015b*). The 18 *PmHsf* genes were widely distributed on chromosomes 2–8, except for *Hsf18*, which was located at scaffold272. Similarly, the *AtHsf* genes are distributed on all five chromosomes, while chromosomes 11 and 12 of the rice genome lack *OsHsf* genes. The finding that *Hsfs* are distributed on almost every chromosome suggests that the *Hsf* genes may have been widely distributed in the genome of the common ancestor of both monocots and eudicots (*Guo et al., 2008*).

Despite considerable differences in the sizes and sequences of the PmHsfs, their basic structure is conserved. Most members of the same subclass exhibited similar motifs or

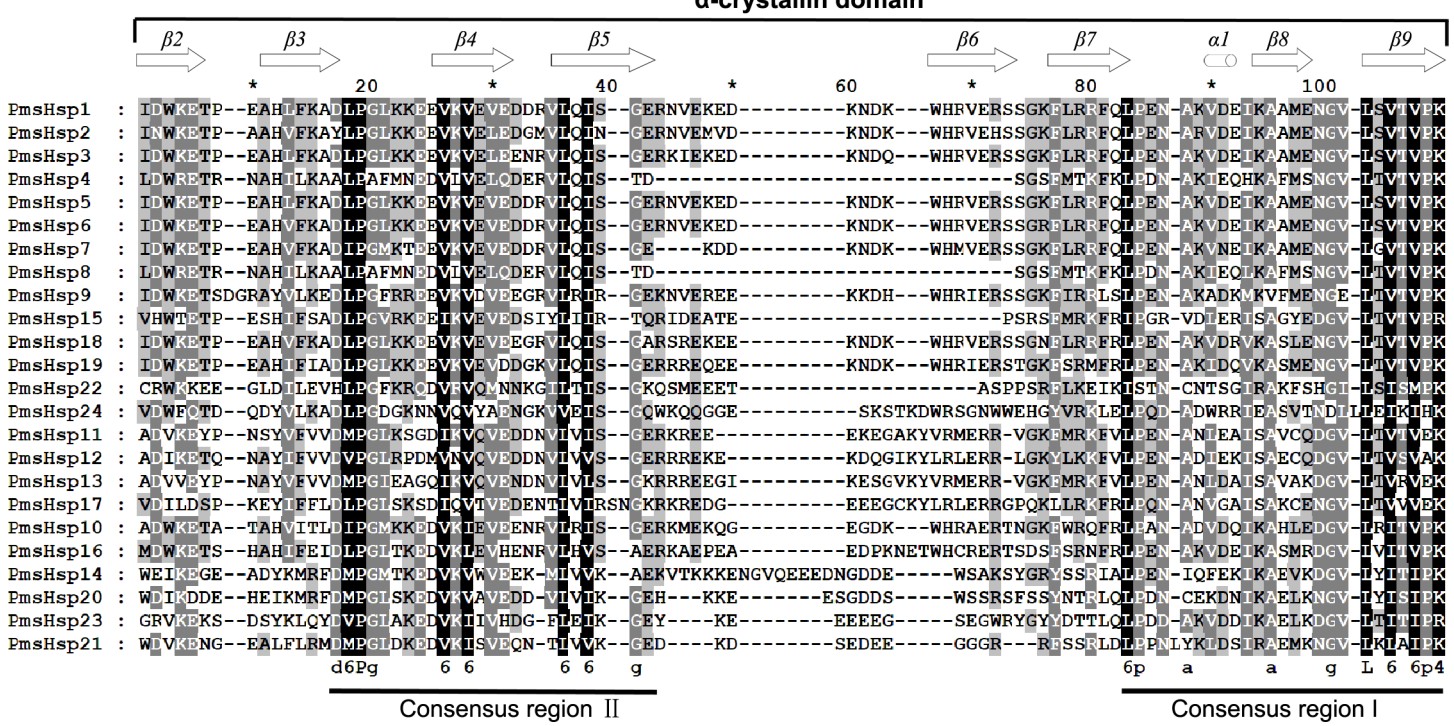

**Figure 6 Multiple sequence alignment of the α-crystallin domain of the sHsp proteins in *P. mume*.** The α-crystallin domain (ACD) comprises two homologous regions, consensus region II and I, separated by a linker, which is more variable in sequence and size. Potential positions of secondary structural elements (β2-β3-β4-β5-β6-β7-α1-β8-β9) are indicated above the alignment.

domain structures compared with AtHsfs. The DBD, consisting of a three-helix bundle and a four-stranded antiparallel β-sheet, was the most conserved element of the PmHsfs. The result was consistent with the AtHsfs (*Nover et al., 2001*). Moreover, the DBD of PmHsfs was encoded in two parts separated by only one intron, and the position of the intron was consistent in all cases, although its size varied greatly (Fig. 2; Table S2); this is consistent with the DBD of AtHsfs (*Nover et al., 2001*). PmHsf2 (A2) contained two introns, which is the same condition as is found in GhHsf2 and GhHsf12 (*Wang et al., 2014*). The HR-A/B region is connected to the DBD by a flexible linker of variable length (15–80 amino acid residues) (*Scharf, Siddique & Vierling, 2001*). A similar pattern is found in *P. mume*. A highly conserved repressor tetrapeptide motif -LFGV- was found in class B Hsfs. Class A PmHsfs were rich in F, W, D, and L amino acid residues. Similar AHA motifs of aromatic residues in an acidic surrounding were identified in *Arabidopsis* (*Nover et al., 2001*; *Scharf, Siddique & Vierling, 2001*).

Small Hsp family proteins confer tolerance to environmental stresses due to their abundance and diversity. In *Arabidopsis* and rice, 19 and 23 *sHsps* have been identified, respectively (*Scharf, Siddique & Vierling, 2001*; *Sarkar, Kim & Grover, 2009*). However, little is known about this family in *P. mume*. In our study, 24 *PmsHsp* genes were identified, of which nine were detected on chromosome 1 (Fig. 1B). Moreover, two clusters of tandem-repeated *sHsp* genes belonging to CI were detected on chromosome 1. Each cluster included three members (*PmsHsp1*, *PmsHsp2*, and *PmsHsp3*; *PmsHsp5*, *PmsHsp6*,

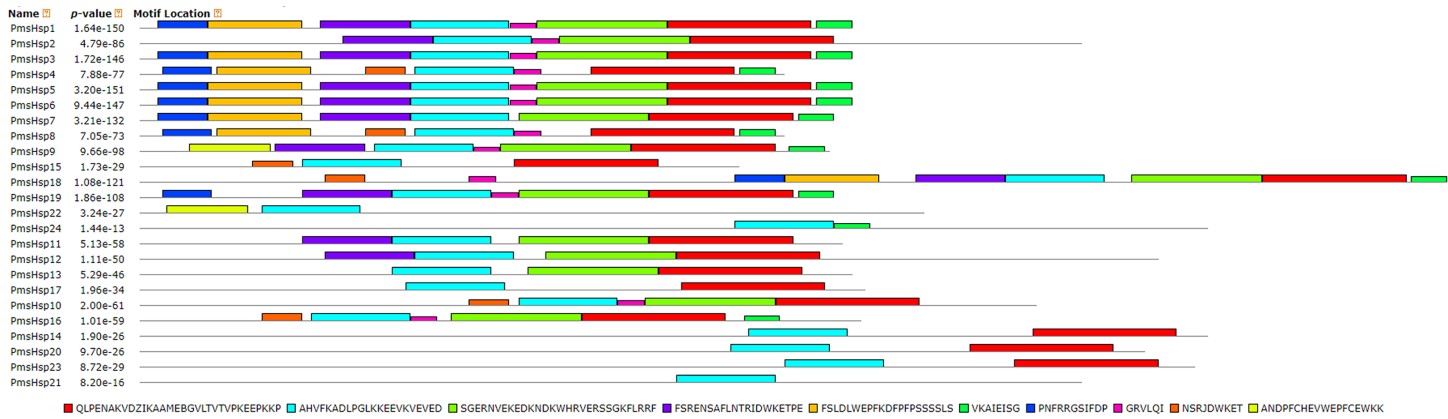

**Figure 7 Distribution of conserved motifs in the *sHsp* family members in *P. mume*.** All motifs were identified by MEME 5.0.5 using the complete amino acid sequences. Motif sizes are indicated at the bottom of the figure. Different motifs are indicated by different colors numbered 1–10. For details of motifs refer to Table 3.

**Table 3 sHsp motif sequences identified in *P. mume* by MEME tools.**

| Motif | Multilevel consensus sequence |
| --- | --- |
| 1 | QLPENAKVDZIKAAMEBGVLTVTVPKEEPKKP |
| 2 | AHVFKADLPGLKKEEVKVEVED |
| 3 | SGERNVEKEDKNDKWHRVERSSGKFLRRF |
| 4 | FSRENSAFLNTRIDWKETPE |
| 5 | FSLDLWEPFKDFPFPSSSSLS |
| 6 | VKAIEISG |
| 7 | PNFRRGSIFDP |
| 8 | GRVLQI |
| 9 | NSRJDWKET |
| 10 | ANDPFCHEVWEPFCEWKK |

**Note:**
Numbers correspond to the motifs described in Fig. 7.

and *PmsHsp7*) (Fig. 1B). Similar *sHsp* clusters were found in rice (*Guan et al., 2004*; *Hu, Hu & Han, 2009*). The present findings may provide a basis for further study of the sHsp family and help to enable the identification of candidate genes that are useful for the breeding of ornamental plants that are responsive to abiotic stress conditions.

As in *Arabidopsis*, PmsHsps shared the considerably conserved ACD, but some differences existed in this domain among the PmsHsp subfamilies (Figs. 6 and 7). For instance, PmsHsp4, PmsHsp8, PmsHsp15, and PmsHsp22 lack the crucial β6 sheet, which is vital for formation and oligomerization of the dimer (*Van Montfort et al., 2001*).

Using phylogenetic analyses of 19 AtsHsp members and 24 PmsHsp members, eight distinct clusters, CI, CII, CIII, ER, P, M, CI(r), and P(r), comprising eight, three, one, two, two, one, six, and one *PmsHsp* genes, respectively, were identified and displayed (Fig. 3). The N-terminal regions of the CI PmsHsps had a WD/EPF domain. Moreover, an interesting discovery was revealed in the CII subfamily (PmsHsp11, PmsHsp12, and

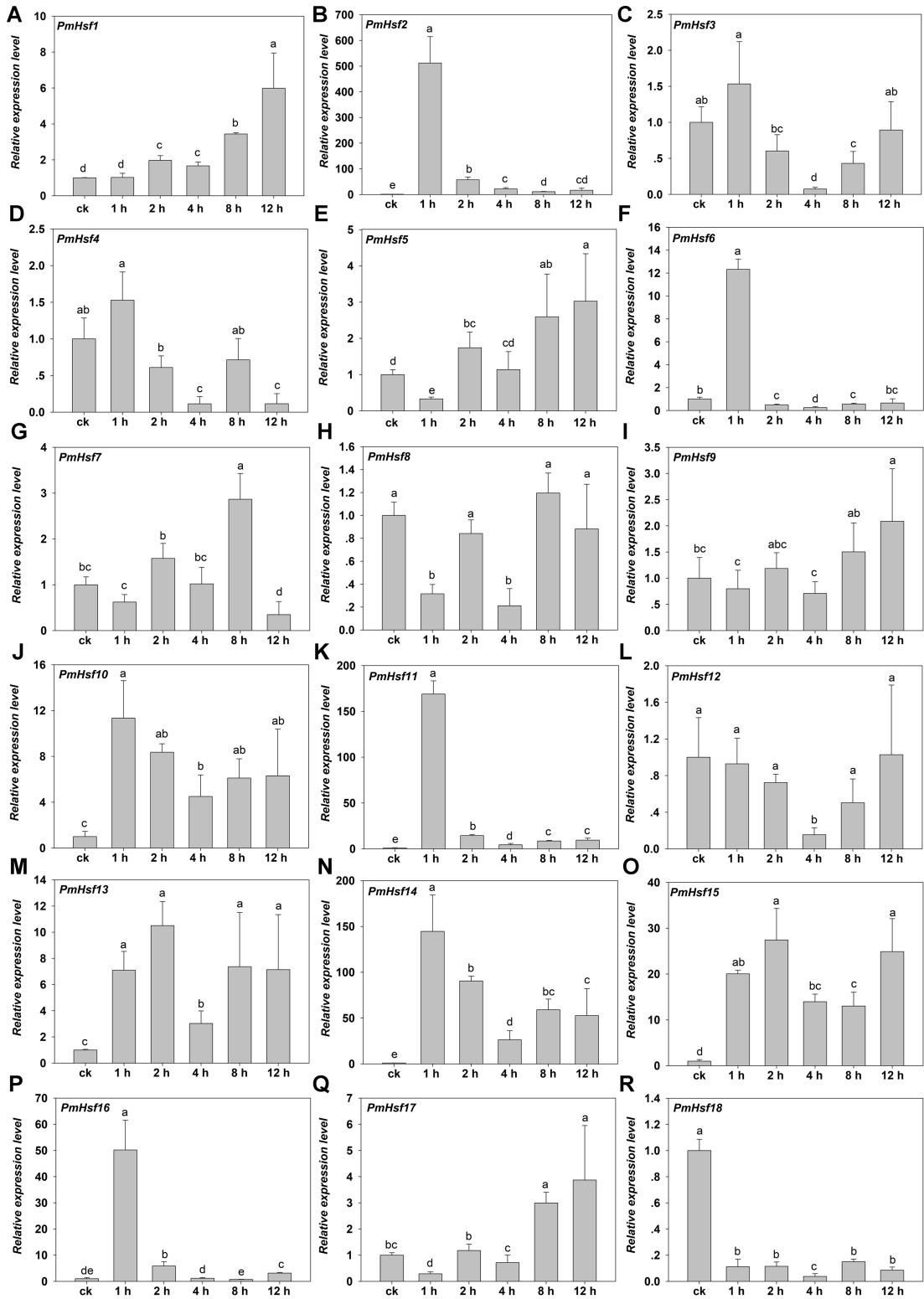

**Figure 8 Relative gene expression of *PmHsf* genes analyzed by qRT-PCR responsed to heat stress treatment.** (A–R) The relative gene expression of *PmHsf1-18*. The vertical axis represents the fold change in expression relative to ck (onefold). The horizontal axis represents the different times of heat stress treatment. Results were normalized using *PmEF1α* gene expression as the reference gene. Error bars indicate standard deviation of three replicates. Different letters indicate that the means are significantly different based on Duncan's test ($p < 0.05$) after one-way ANOVA.

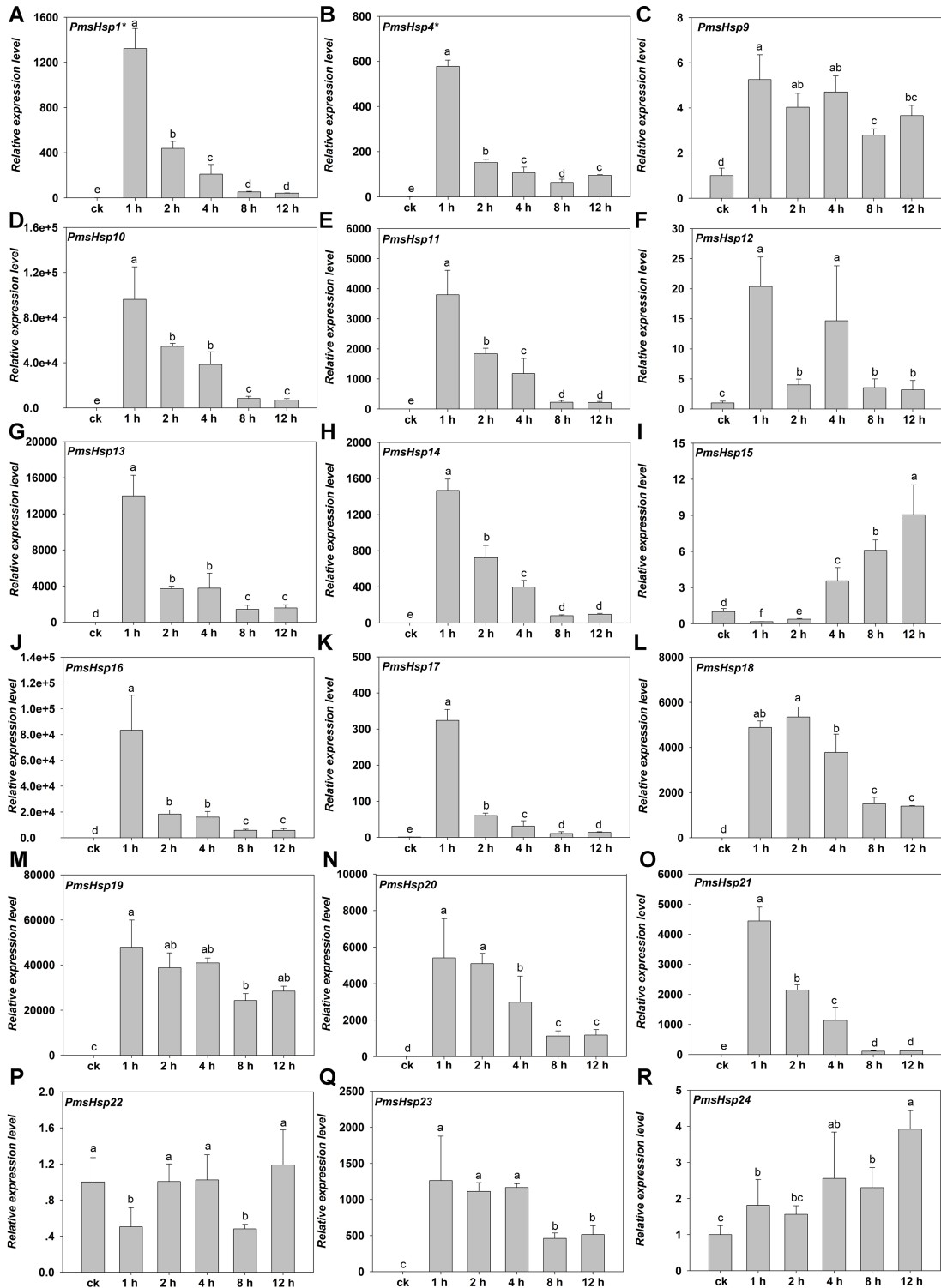

**Figure 9 Relative gene expression of *PmsHsp* genes analyzed by qRT-PCR responsed to heat stress treatment.** (A–R) The relative gene expression of *PmsHsp1-24*. The vertical axis represents the fold change in expression relative to ck (onefold). The horizontal axis represents the different times of heat stress treatment. Results were normalized using *PmEF1α* gene expression as the reference gene. Error bars indicate standard deviation of three replicates. Different letters indicate that the means are significantly different based on Duncan's test (*p* < 0.05) after one-way ANOVA. *sHsp1*[*] represent the total relative expression level of *sHsp1*, *sHsp2*, *sHsp3*, *sHsp5*, *sHsp6*, and *sHsp7*. *sHsp4*[*] represent the total relative expression level of *sHsp4* and *sHsp8*.

PmsHsp13) (Fig. S2), which had a conservative N-terminal amino acid motif (DA-AMAATP) that was not detected in the other cytoplasmic/nuclear sHsps (*Waters, 1995*).

## Expression patterns of *Hsfs* and *sHsps*

Heat shock transcription factors are the major regulators of HSR in plants. qRT-PCR analysis revealed that *PmHsf* genes were differentially expressed under HS (Fig. 8). *PmHsf18* (B3) was significantly inhibited under HS. *PmHsf8* and *PmHsf12*, belonging to the *HsfA1* subfamily, were not up-regulated under HS, similar to the expression of the *HsfA1* genes in *Arabidopsis*, rice, and wheat (*Hübel & Schöffl, 1994*; *Mittal et al., 2009*; *Xue et al., 2014*). Among the genes with significantly increased expression, *PmHsf2* (A2), *PmHsf11* (A7), and *PmHsf14* (A3) were early HSR genes and became predominant transcripts during HS, especially *PmHsf2* (A2), which had a transcript level that was enhanced more than 500-fold 1 h after HS treatment (Fig. 8). This result was consistent with the genome-wide analyses of *Hsfs* in maize and cotton (*Lin et al., 2011*; *Wang et al., 2014*). Class B members of the PmHsf family (*PmHsf1*, *PmHsf6*, *PmHsf10*, *PmHsf13*, and *PmHsf15*) appeared to be induced by HS, with the exception of class B3 (e.g., *PmHsf18*), which was inhibited during HS. In class C, *PmHsf17* was up-regulated and showed the highest expression level at 12 h.

In agreement with the previous findings that most of plant sHsps are highly up-regulated during HS, 23 of 24 *PmsHsps* exhibited variable degrees of up-regulation (Fig. 9), indicating that sHsps may play general roles in adaptation to HS. There was no single, specific expression pattern for the CI and CII subfamilies during HS; however, the ER, M, and P subfamilies shared a similar expression pattern. These same sHsp expression patterns were observed in rice under heat treatment (*Sarkar, Kim & Grover, 2009*).

## CONCLUSIONS

This study presents a comprehensive overview of the genomic complexity and expression diversity of 18 *Hsfs* and 24 *sHsps* from *P. mume*. Structural characteristics and phylogenetic analyses revealed divergent expansion patterns of the *Hsf* and *sHsp* gene families in different classes and subclasses. Furthermore, qRT-PCR analysis revealed that most of the *Hsf* and *sHsp* genes were highly up-regulated in response to high temperature.

## ACKNOWLEDGEMENTS

We thank all of our colleagues in our laboratory for their constructive discussions and technical support.

### Funding

This work was supported by the National Key R&D Program of China (No. 2018YFD1000402) and the National Natural Science Foundation of China (No. 31270739). The funders had no role in study design, data collection and analysis, decision to publish, or preparation of the manuscript.

## Grant Disclosures

The following grant information was disclosed by the authors:

National Key R&D Program of China: 2018YFD1000402.

National Natural Science Foundation of China: 31270739.

## Competing Interests

The authors declare that they have no competing interests.

## Author Contributions

- Xueli Wan performed the experiments, analyzed the data, prepared figures and/or tables, approved the final draft.
- Jie Yang performed the experiments, approved the final draft.
- Cong Guo conceived and designed the experiments, contributed reagents/materials/analysis tools, approved the final draft.
- Manzhu Bao authored or reviewed drafts of the paper, approved the final draft.
- Junwei Zhang conceived and designed the experiments, approved the final draft.

## Data Availability

The raw data are available in the Supplemental Files. The raw data shows the amino acid sequence, coding sequence, gDNA sequence and relative gene expression of PmHsf and PmsHsp genes.

## Supplemental Information

Supplemental information for this article can be found online at http://dx.doi.org/10.7717/peerj.7312#supplemental-information.

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
