# Peer review of "Genome-wide identification and classification of the Hsf and sHsp gene families in Prunus mume, and transcriptional analysis under heat stress"

_PeerJ, doi:10.7717/peerj.7312_

## Round 0.1 · original submission · Major Revisions

Based on the comments from the reviewers, please make revision on this manuscript.

·

Basic reporting

The manuscript requires some further editing to correct grammar throughout.

Experimental design

The introduction provides a general review of plant Hsfs and Hsps, but it is not clear how this material sets up the rationale for this study, or the gaps in knowledge that this study fills. The data provided in this study are clearly unique, but I think the authors could do a better job of explaining how their work connects to and expands on what is known about plant Hsfs and Hsps. The authors could also be clearer on why this study focuses on the small Hsps specifically. How is it beneficial to add information on these genes for P. mume?

Phylogenetic analysis: I suggest describing the settings used in Mega to produce the phylogenies. If the default settings were used that should be stated.

qRT-PCR design: There are a number of details that I think are important to include in the methods section. It would be helpful if the authors stated whether primers were designed to produce products spanning introns to reduce the chance that genomic DNA was amplified (for genes that contain introns). Were products from each primer pair sequenced to determine if the correct gene was amplified? Was melt curve analysis used to ensure a single product was generated? Were negative RT and water negative controls used? Has EF1a been validated as a good reference gene for P. mume or at least other plant qRT-PCR experiments? If not, why was this gene chosen? What range in Cq value was accepted within a single technical triplicate (0.5 or other value)? The use of two reference genes would improve the strength of these data, and could be considered for any future experiments.

I would suggest some other revisions to the description of the qRT-PCR design as recommended by the MIQE guidelines. These suggest referring to control genes as “reference genes” and the use of Cq instead of Ct. Were efficiencies calculated for each primer pair?

Validity of the findings

The data in the paper are well presented and described. Figures are effective. Conclusions resulting from the figures are valid. The descriptions of upregulation and downregulation in the qRT-PCR data seem valid as the changes in expression are so large for many of the genes. It is a bit confusing that each y-axis differs depending on the amount of gene expression change, which visually makes it difficult to compare the extent of expression change from gene to gene. Perhaps the authors can clearly note that y-axis scales differ to allow presentation of all data in one figure. It might also be helpful to clearly state the fold difference that was considered significant. Is it appropriate to use an ANOVA to show statistical significance for these changes?

Most of the first paragraph of the discussion reiterates information from the introduction. I would recommend using this first paragraph to stress the contributions of this current study. Line 251 beginning paragraph 2 could more clearly state that it is similarities between P. mume and Arabidopsis that was conserved.

Paragraph 3 starting on line 258 seems to restate some of the findings, but could better explain the importance of these findings.

Paragraph 4 restates a result. Is this loss of the beta 6 sheet found in any other plants or eukaryotes? Or is this a unique finding in P. mume?

Paragraph 5 starting on line 270 restates information from the introduction on cellular localization. Since no cellular localization data are provided for expression of the P. mume genes I am not sure if this information is relevant in the discussion.

The discussion on timing of gene expression during heat stress also restates the results. How do these results compare to what is known about timing of heat shock response in other eukaryotes? What do these data add to our understanding of heat shock response in plants? Why might these different temporal patterns exist?

Additional comments

The authors present a novel data set that will be of interest to researchers working on plant heat stress response. The data appear valid and well presented. Most of my comments, detailed in the other sections of this review, relate to the written presentation of the paper and the design and analysis of the qRT-PCR experiment.

Here are a few other minor comments:

Abstract – “Heat” in first sentence should probably be lower case. Authors could clarify what they mean by “outstanding features” in the second sentence. While heat shock factors (HSPs) is defined in the abstract, sHSP is not. I would define that “s” stands for “small”. When authors state that certain Hsfs became “prominent”, under what conditions were they prominent? I assume this means after heat shock, but that could be clarified. “PmHsp” should not be italicized.

Line 33: Change to “relieved”

Line 128: Change to RNase and DNase

Line 214: PmHsf and PmsHsp does not refer to specific genes in this sentence, so I do not think they should be italicized.

·

Basic reporting

The English language should be improved to ensure that an international audience can clearly understand your text. Some examples where the language could be improved include lines line 37, 148-155, 156, 159-162, 163-165, 259-260 and 282-284. – the current phrasing makes comprehension difficult. I would suggest writing short sentences and asking a native English-speaking reader to read the manuscript before submission.

Line 176 – 177 - The length of the introns varies among the Hsf genes – for example, PmHsfX and PmHsf4 have smallest (?bp) and longest (3196bp) intron respectively.

Line 282: exist should be exhibit

Specifying the acronym small HSP (sHSP) in the abstract (instead of line 51) to help the readers to understand it early.

Fig 1 A: Because the chromosome start and end is clear from the scale, it is not required to mention with each chromosome. For each chromosome please mention the position of the centromere.

Experimental design

The Prunus mume genome database (http://prunusmumegenome.bjfu.edu.cn/) is not accessible. That creates difficulty to run some analysis by myself.

Line 166 – It is not clear why there is a difference between phylogenetic analysis and annotation? What are the (de)merits of this information?

The figure legend of fig 8 and 9 is telling there are 3 biological replicates, however, line 138 (method section) suggesting there are three technical replicates. Moreover, line 124 is suggesting there are three biological replicates. Authors need to clarify at one place how many biological replicates and technical replicates are used in the qPCR experiment.

Validity of the findings

The discussion section is not written in a very thoughtfully way. There are lots of redundancy between the discussion and the introduction section. It is more focused on other plant’s Hsf and sHsp (i.e. what we already know) rather than what new things we are learning for P. mume. This section should include, what is the significance of the current study to improve/ help in conservation of P. mume.

Reviewer 3 ·

Basic reporting

The manuscript “Genome-wide identification and classification of Hsf and sHsp gene families in Prunus mume, and their transcriptional analysis under heat stress” attempts to provide structural as well as functional characterization of Hsfs and sHsps in P. mume. Overall, the manuscript conforms to the structure recommended by the journal viz. Introduction, Materials and Methods, Results, Discussion, Conclusions. I found the manuscript very well written. The overall English comprehension and grammar of the article are good with very few typos and ambiguous sentences. Figures are of high quality. The methods and results section is explained in satisfactory detail. However, there are few suggestions that would be useful for the readers and are important to address to make the manuscript scientifically thorough. More specific suggestions include:

1. In line 3 of the abstract, please be specific about the “outstanding features of Prunus mume”.
2. I suggest to combine paragraph 1 and 2 of the introduction as a single paragraph as both talks about Hsfs and demands continuity.
3. An ambiguous sentence in line 32. Please rephrase.
4. Incorrect grammar in the sentence at line 61.
5. In line 90, the sentence is confusing such that if the author has done genome sequencing of Prunus mume or was it done already? Please rephrase.
6. In line 113, please correct neighbor-joining (NJ) methods to “neighbor-joining (NJ) method” as it is one method.
7. Remove extra comma in line 186 in “(Fig. S1,)”
8. P. mume is not italicized in Table 1 legend.
9. In the legend of Table 2, “MEME tools” should be singular.

Experimental design

1. In the introduction, I suggest the author to rephrase the sentences from line 81 - 87 to be in active voice. For e.g.” In this study, we performed genome-wide identification of Hsf and sHsp ……” etc.
2. In the “Methods” section, please indicate the version of all the software used for the analysis, wherever applicable. For e.g. MEME, MEGA etc.
3. In MEME analysis, authors have reported motifs with different length. Please include all the parameters of MEME analysis in the method section.
4. A detailed characterization of Hsf and sHsp in P. mume is missing. In particular, the author can provide a figure with distribution of Hsf and sHsp in P. mume among different classes based on DNA binding domains and sub-cellular localization.
5. Author has provided a comparative analysis of P. mume with Arabidopsis. In addition, a bar chart indicating the number of Hsf and sHsp in different species (if possible, for different category) would be very informative in the context of the present study. It would help readers to understand the discussion section that also includes below lines - “The number of Hsf genes in pear and apple was nearly twice as much like that in peach, strawberry, and Mei flower, which may be the result of the whole genome duplication (WGD) event in the Maloideae. Similar situation was observed in Populus trichocarpa, Zea mays (Lin et al., 2011; Zhang et al., 2015)”.
6. Is there any class of Hsf and sHsp that is more prominent or specific to P. mume or similar lineage? Please comment.
7. The result section is written using passive voice. I would suggest the authors read the result section and rephrase the sentences with the active voice to present the findings of the study in an exciting and interesting manner.

Validity of the findings

The study is experimentally sound and well presented. For further comments refer section Experimental design.

---

## Round 0.2 · Minor Revisions

Dear Dr. Zhang,
I have received comments on your manuscript from two reviewers. Please make changes according to the comments from reviewer 1.

·

Basic reporting

The authors have made edits to address my comments and the grammatical issues throughout appear to have been addressed. The introduction does a better job of setting up the rationale and importance of the study.

My comments about reporting in the discussion have also been addressed.

Experimental design

The authors have responded to my comments and added details about their phylogenetic and qRT-PCR analysis.

The authors addressed my questions about their qPCR design in their rebuttal letter, but I think some of this information should also go in the Methods section. These additions would help address PeerJ’s guidelines that qPCR analyses follow the MIQE guidelines.

Specifically, I recommend stating that each qPCR primer pair was tested by end-point PCR to show that they produced a specific product of predicted size. But I would also add that these products were not sequenced to confirm their identity. While I don’t necessarily suggest that this work be repeated to sequence each product, I would strongly recommend that this be done in future studies. I also recommend that the authors state that primer efficiencies were not calculated, and suggest that this be done in future studies, and that melt curve analysis be completed after each reaction.

I also recommend stating what controls were done. It appears that water non-template controls were used, but not negative RT controls. This should be made clear in the Methods section. It would also be helpful to add the citation supporting the use of the reference gene and to state what Cq variation was acceptable between technical triplicates.

Validity of the findings

The authors have added a statistical analysis of gene expression differences.

Additional comments

I appreciate the time taken by the authors to revise their manuscript and address my suggestions. The manuscript is much improved. I believe that there are still some needed changes to the description of the qPCR analysis.

---

## Round 0.3 · accepted · Accept

After careful consideration, your manuscript is acceptable for publication.

·

Basic reporting

no comment

Experimental design

no comment

Validity of the findings

no comment

Additional comments

The authors have addressed my final suggestions.